# A Study on the Improvement of the Fatigue Life of Bearings by Ultrasonic Nanocrystal Surface Modification Technology

**Shirmendagva Darisuren [1], Jeong-Hyeon Park [2], Young-Sik Pyun [2] and Auezhan Amanov [2,\*]**

[1] Graduate School, Department of Mechanical Engineering, Sun Moon University, Chungcheongnam-do 31460, Korea; shirmee9999@naver.com

[2] Department of Mechanical Engineering, Sun Moon University, Chungcheongnam-do 31460, Korea; pjh@sunmoon.ac.kr (J.-H.P.); pyoun@sunmoon.ac.kr (Y.-S.P.)

\* Correspondence: avaz2662@sunmoon.ac.kr; Tel.: +82-41-530-2892

**Abstract:** In this study, the effects of ultrasonic nanocrystal surface modification (UNSM) technology on the fatigue life of needle roller bearings were investigated. The fatigue life of the untreated and UNSM-treated needle roller bearings was evaluated using a roller fatigue tester at various contact stress levels, under oil lubrication conditions. It was found that the fatigue life of the UNSM-treated needle roller bearing was extended by approximately 34.3% in comparison with the untreated one. The results of the surface roughness and surface hardness of the needle roller bearings before and after UNSM technology were compared and discussed in order to understand the role of UNSM technology in improving fatigue life. It was found that the application of UNSM technology to the needle roller bearings can improve their fatigue life by reducing the friction coefficient and increasing the wear resistance, which may be attributed to the reduction in surface roughness from 0.50 μm to 0.15 μm and also the increase in surface hardness from 58 HRC to 62 HRC.

**Keywords:** needle roller bearing; surface roughness; surface hardness; rolling contact fatigue; ultrasonic nanocrystal surface modification

---

## 1. Introduction

A bearing is one of the important mechanical components and is indispensable in almost all industries, such as steel, automotive, power plant equipment, and aerospace. [1]. In particular, the usage of bearings in excavator speed reducers, large rolling mills, aircraft and ships requires a special bearing with a high-level design and durability. The fatigue life of bearings is usually indicated by the number of revolutions until flaking occurs on the raceway surface, due to the fatigue caused by the cyclic stress on the raceway surface during operation [2,3]. A number of investigations concerning material design, machining, heat treatment and surface treatment have been conducted to improve dynamic load rating/static load rating, which is the indication of the strength and durability of the bearings [4,5].

In this study, the effect of ultrasonic nanocrystal surface modification (UNSM) technology on the fatigue life of needle roller and angular ball bearings was investigated. These bearings are widely used in rear wheel drive transmission, constant velocity joints and mechanical pumps. Surface and microstructure-related damage such as flaking, cracking and wear, is caused by friction and fretting during the operation and can affect the fatigue life of bearings [6]. In this regard, it is necessary to improve the fatigue life of bearings by controlling the surface properties of bearing raceways. A UNSM technology, which was developed in South Korea, is applied to the surface of the inner and outer raceways of bearings to improve the fatigue life by reducing the surface roughness, increasing the strength, and transferring a tensile residual stress into compressive residual stress.

The novelty of this study is the effect of UNSM technology that has shed light on the fatigue life fracture mechanisms of bearings. The main objective of this study is to find out the fatigue life extension of bearings by the application of UNSM technology. Even though many researchers have worked on the fatigue life of bearings, very few researchers have reported about the effects of UNSM technology on the fatigue life that is the main criterion for designing efficient and reliable bearings. Two different bearings which are used in automobiles were taken into consideration and it was found that the fatigue life of the UNSM-treated bearings is longer than the commercially available bearings. It is believed that the findings from this study are very useful for the design of bearings and will aid in improving their durability and reliability. Hence, the results of this investigation are expected to make these UNSM-treated bearings more attractive for a range of applications in various industries.

## 2. Needle Roller Bearing Fatigue Life Test

In this study, both the ring specimens, with a diameter of 30 mm and thickness of 8 mm, and the roller specimens with a diameter of 2 mm and thickness of 4.5 mm, made of SAE52100 bearing steel were prepared for the rotary contact fatigue (RCF) (SMU, Asan, Korea) test of needle roller bearings. The RCF test configuration is shown in Figure 1, where the specimen-A ring was a fixed body, while the specimen-B ring was a rotating body. The RCF tests of needle roller bearings were performed using 14 rollers, which came into contact with the specimens in rolling motion once a load was applied. A flowchart of the research stages is presented in Figure 2 where the specimens were classified first; after which, some of them were treated by UNSM treatment, then the surface properties of the untreated and UNSM-treated specimens were measured and characterized. Afterwards, once the expected surface properties were obtained by UNSM treatment, the specimens were subjected to fatigue tests in case there were no improvements in surface properties after UNSM treatment. Then, the specimens were re-treated by UNSM technology under different parameters.

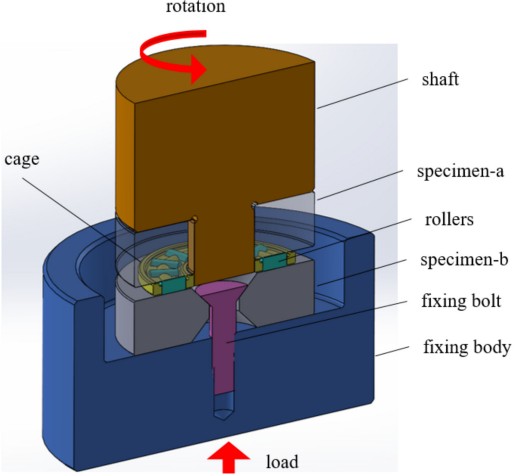

**Figure 1.** Needle roller bearing fatigue test configuration.

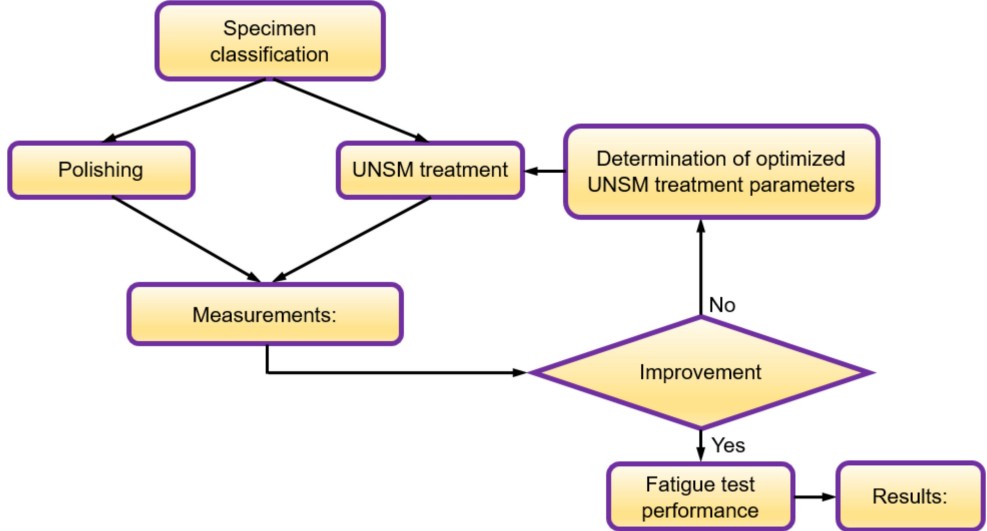

**Figure 2.** A flowchart describing the research stages.

### 2.1. Ultrasonic Nanocrystal Surface Modification (UNSM) Technology

UNSM technology is one of the surface modification technologies which utilizes an ultrasonic energy. In UNSM technology, a tungsten carbide ball is attached to an ultrasonic horn that strikes the surface with a frequency of 20 kHz as shown in Figure 3a. The detailed description of this technology was described in our previous study [7]. Strikes, which can also be described as micro cold forging, cause severe plastic and elastic deformations on the surface layers, thus inducing a nanocrystalline structure and deep compressive residual stress [8]. These strikes also produce controllable micro-dimples on the top surface of the specimen, which improves the tribological characteristics of the interacting surfaces in relative motion. Each of these micro-dimples can serve as a hydrodynamic bearing in cases of full or mixed lubrication, or a lubrication reservoir for lubricants in cases of starved lubrication conditions, or wear debris in lubricated sliding/rolling conditions [9]. The produced nanostructure of the surface layer after UNSM treatment can simultaneously improve the strength (hardness) and ductility (toughness) of the specimen, according to the well-known Hall-Petch relationship [10]. Figure 3b shows the UNSM treatment processing under the parameters listed in Table 1.

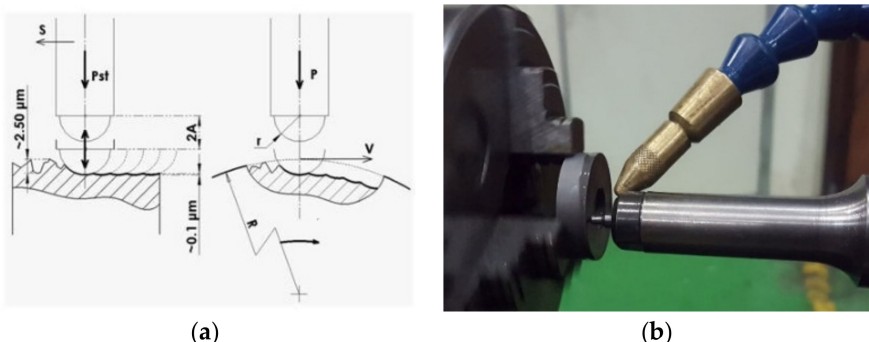

| (a) | (b) |

**Figure 3.** (**a**) Ultrasonic nanocrystal surface modification (UNSM) treatment principles and (**b**) the processing of the specimen.

**Table 1.** UNSM treatment process.

| Amplitude (µm) | Load (N) | Rotational Speed (rpm) | Feed (mm/rev) |
| --- | --- | --- | --- |
| 30 | 60 | 60 | 0.07 |

### 2.2. Fatigue Test Conditions

The fatigue life test was conducted at several contact stress levels using the UNSM-treated and untreated specimens to investigate the effect of UNSM treatment on the fatigue life of bearings. The fatigue tests were repeated two times to obtain reliable data. The maximum contact stress along the centerline of the rectangular contact area was calculated by Equation (1):

$$P_{max} = \frac{2F}{\pi b L} \tag{1}$$

The value of b of the rectangular contact area of two parallel cylinders was calculated by Equation (2):

$$b = \sqrt{\frac{4F\left[\frac{1-v_1^2}{E_1} + \frac{1-v_2^2}{E_2}\right]}{\pi L\left(\frac{1}{R_1} + \frac{1}{R_2}\right)}} \tag{2}$$

Calculated contact stresses for different specimens used in the test are listed in Table 2, while the fatigue life test conditions are listed in Table 3.

**Table 2.** Specimen type and contact stress.

| Specimen Types | Contact Stress (Mpa) |
|---|---|
| #1NRB-untreated<br>#2NRB-UNSM treatment | 1770 |
| #3NRB-untreated<br>#4NRB-UNSM treatment | 1715 |
| #5NRB-untreated<br>#6NRB-UNSM treatment | 1600 |
| #7NRB-untreated<br>#8NRB-UNSM treatment | 1484 |
| #9NRB-untreated<br>#10NRB-UNSM treatment | 1355 |

**Table 3.** Fatigue life test conditions.

| Rotational Speed (rpm) | Lubrication Conditions | Vibration Fatigue Setting Value |
|---|---|---|
| 1500 | Oil | 22 |

### 2.3. Effects of UNSM Treatment

Figure 4 shows the scanning electron microscope (SEM: JSM-IT200, Jeol, Tokyo, Japan) images of the untreated and UNSM-treated specimens. Figure 4a shows the polished surface with uneven grooves that may lead to a reduction in the fatigue life of bearings. In Figure 4b, it was confirmed that the uneven grooves were eliminated and micro-dimples were formed after UNSM treatment. It has already been proven that the micro-dimples can improve the fatigue life of bearings with the presence of oil-film thickness between the mating surfaces in relative motion [6].

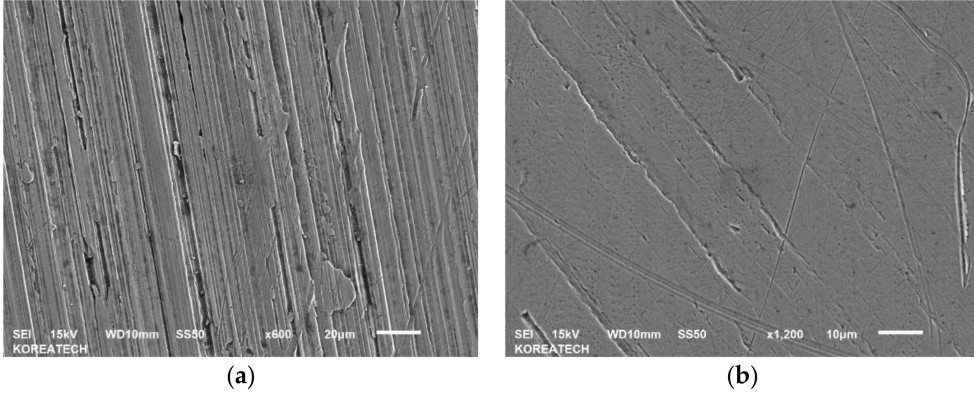

**Figure 4.** SEM images of the untreated (**a**) and UNSM-treated (**b**) specimens.

Table 4 shows the comparison in surface roughness and hardness of the specimens before and after UNSM treatment. The surface roughness of the UNSM-treated specimen-A was reduced from 0.550 μm to 0.149 μm, while the surface hardness was increased from 58 HRC to 62 HRC. Hence, it is believed that the friction behavior, wear resistance and the fatigue life of bearings can be improved.

**Table 4.** Comparison of the roughness and hardness of the specimens before and after UNSM treatment.

| Specimen | Roughness, (μm) | | Hardness, (HRC) | |
|---|---|---|---|---|
| Comparison | Before | After | Before | After |
| Specimen-A | 0.550 | 0.149 | 58 | 62 |
| Specimen-B | 0.477 | 0.150 | 58 | 62 |

Stresstech XSTRESS 3000 X-ray diffraction (XRD) was used to measure the residual stress of the untreated and UNSM-treated specimens. The measurement conditions were analyzed using a collimator diameter of 2 mm, an exposure time of 20 s, a diameter of 3 mm and an exposure time of 10 s. The residual stress measurement results of the inner ring raceways surface of the untreated and UNSM-treated specimens are listed in Table 5. It was found that the residual stress measured at the surface of the UNSM-treated specimen was increased by 21 times comparative to the untreated specimen. It is obvious that the UNSM treatment induced high and deep compressive residual stress to the surface and sub-surface. Compressive residual stress is an important factor to improve the fatigue life of bearings [11].

**Table 5.** Comparison of the residual stress of the inner bearing raceway before and after UNSM treatment.

| Comparison | Untreated (Mpa) | UNSM-Treated (Mpa) |
|---|---|---|
| Spherical roller bearing inner ring raceway | −54.44 | −1095.75 |

Figure 5 shows the cross-sectional electron backscattered diffraction (EBSD: Tescan MAIA 3 Oxford Instrument Nordylus, Oxford, UK) images of the untreated and UNSM-treated specimens. The EBSD images show that the grains on the top surface of the UNSM-treated specimen were refined into nano-grains with the help of severe plastic deformation [10]. The grain size of the untreated and UNSM-treated specimens was measured by analyzing the EBSD map, using the TSL OIM Analysis 5 program (software for EBSD data acquisition and processing). The refined grain size was found to be in the range of 50 nm to 500 nm from the top surface along with depth. Figure 6 shows that the highest point of surface roughness of the untreated specimen was reduced from 20.3 μm to 17.8 μm after UNSM treatment.

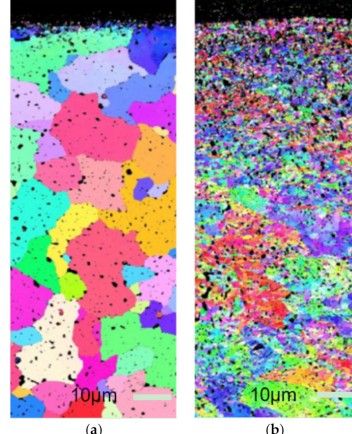

**Figure 5.** Cross-sectional EBSD images of the untreated (**a**) and UNSM-treated (**b**) specimens.

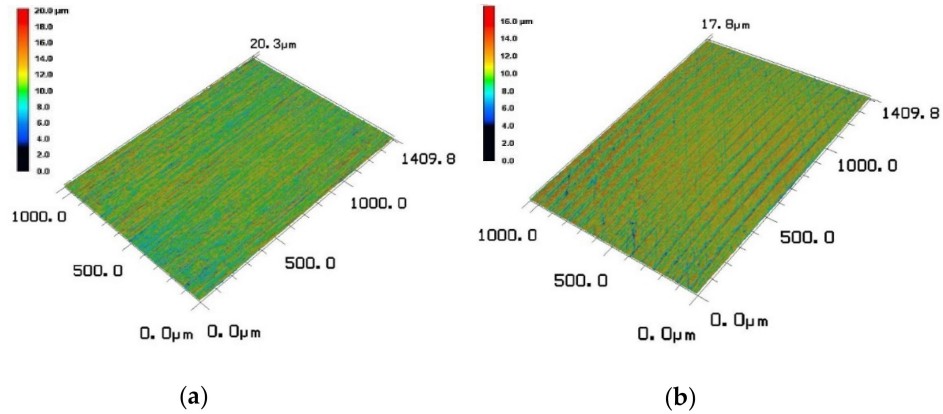

**Figure 6.** 3D LSM images of the untreated (**a**) and UNSM-treated (**b**) specimens.

*2.4. Fatigue Test Results*

Figure 7 represents the S-N curve of the untreated and UNSM-treated specimens derived from fatigue life. The results show the improvement in the fatigue life of the UNSM-treated specimen by 20.5%, 34.3%, and 32.4% at 1770 Mpa, 1715 Mpa, and 1600 Mpa, respectively, compared to the untreated specimen. The longest fatigue life at 1484 Mpa contact stress was found to be a 70.1% increase in the fatigue life of the UNSM-treated bearing compared to the untreated specimen. SEM analysis was performed on the untreated and UNSM-treated specimens to observe a fracture and failure mechanism. The fatigue fractography images of the untreated and UNSM-treated specimens are shown in Figures 8–10. The fractography observation in microstructural change, spalling formation and crack growth when the specimen was subjected to repeated stress in the fatigue life test under the same test conditions are shown in Figures 8 and 9. As shown in Figure 8a,b, it was confirmed that spalling occurred due to repeated fatigue on the untreated #5NRB and UNSM-treated #6NRB specimens. Figure 9a,b show the cross-sectional SEM images of the untreated #5NRB and UNSM-treated #6NRB specimens with a spalling depth of 36.0 μm and of 40.0 μm, respectively. The peeling depth of the untreated specimen showed that a larger exfoliation was produced. Figure 10a,b show the fractography observation of the repeated fatigue life test under the same test conditions. Figure 10a shows the fractography of the untreated #9NRB specimen with a significant spalling. However, Figure 10b shows that the UNSM-treated #10NRB-UNSM specimen had less spalling and cracking. It can be concluded that the UNSM treatment was found to prevent and reduce spalling and cracks that greatly affect the fatigue life improvement of bearings.

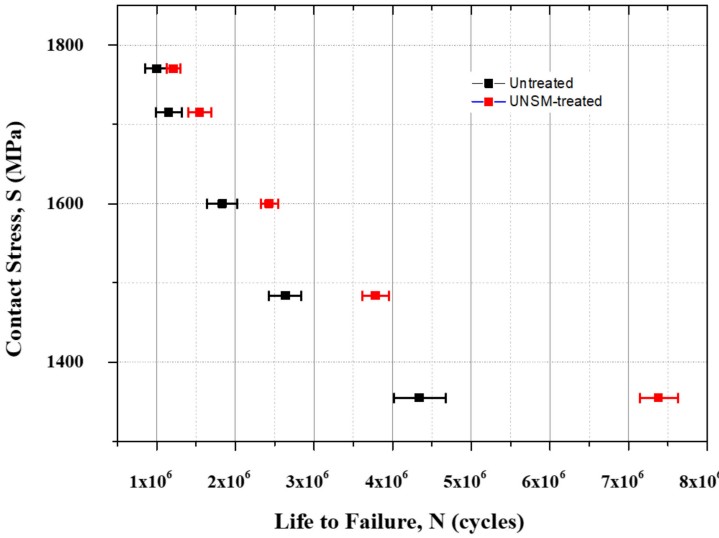

**Figure 7.** Comparison of the S–N curve of the untreated and UNSM-treated specimens.

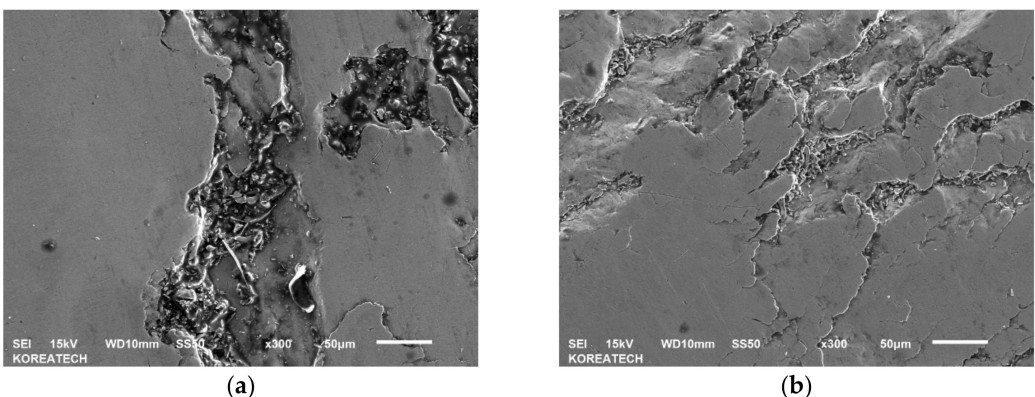

(**a**)　　　　　　　　　　　　　　　　　　　　(**b**)

**Figure 8.** SEM images of the untreated #5NRB (1.150 × 106 cycles) (**a**) and UNSM-treated #6NRB (1.544 × 106 cycles) (**b**) specimens. Testing conditions: Equivalent stress −1600 MPa, rotation speed −1500 rpm.

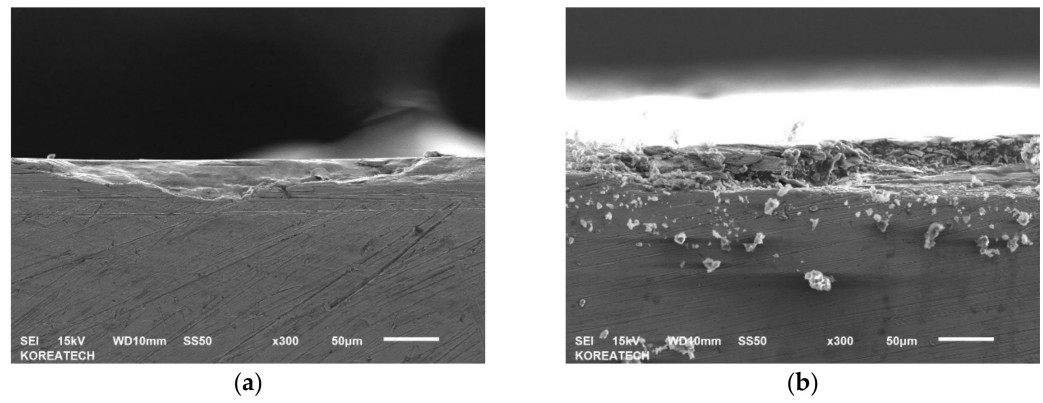

(**a**)　　　　　　　　　　　　　　　　　　　　(**b**)

**Figure 9.** Cross-sectional SEM images of the untreated #5NRB (1.150 × 106 cycles) (**a**) and UNSM-treated #6NRB (1.544 × 106 cycles) (**b**) specimens. Testing conditions: Equivalent stress −1600 MPa, rotation speed −1500 rpm, Fatigue cycles −10.

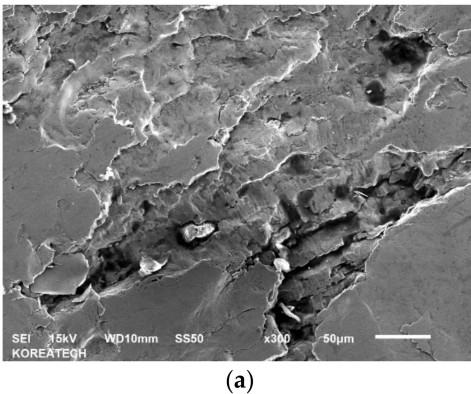
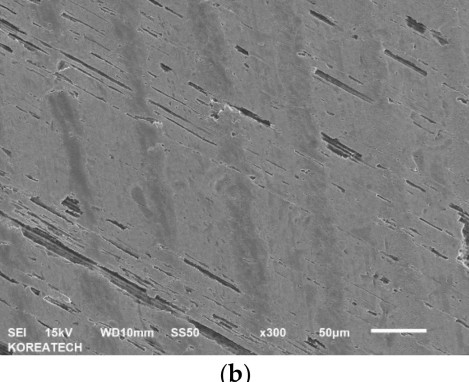

(**a**)                              (**b**)

**Figure 10.** SEM images of the untreated #9NRB (**a**) (4.338 × 106 cycles) and UNSM-treated #10NRB (**b**) (7.380 × 106 cycles) specimens.

## 3. Angular Contact Ball Bearing Fatigue Life Test

### 3.1. UNSM Treatment and Fatigue Test Conditions

The effect of UNSM treatment on the fatigue life of angular contact ball bearings was also investigated. Angular contact ball bearings with a designated number of 7200A were selected as specimens for the fatigue life test. A schematic view of the fatigue test configuration along with dimensions and dynamic load of the angular contact ball bearing are shown in Figure 11 and listed in Table 6, respectively. The inner raceway of the angular contact ball bearing was treated by UNSM treatment under the parameters as listed in Table 7. The fatigue life test of the angular contact ball bearings was performed using the test configuration as shown in Figure 11. During the test, the outer ring was fixed with the shear load on it while the UNSM-treated inner ring rotated. The fatigue life test was conducted in the state that only the shear load was applied to the bearing. The test conditions used in this study are listed in Table 8. It needs to be mentioned here that only the inner raceway was treated with UNSM treatment as it bears more stress compared to the outer raceway.

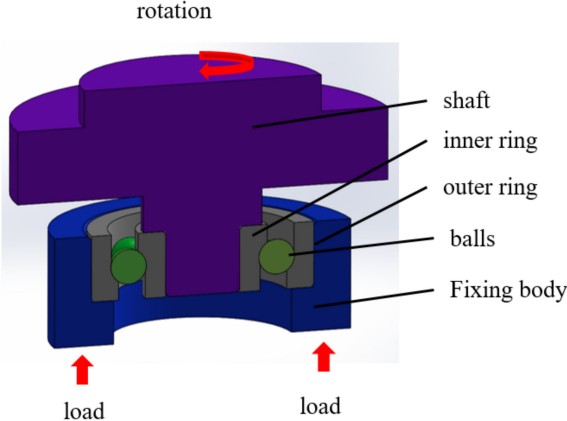

**Figure 11.** Schematic view of an angular contact ball bearing fatigue test configuration.

**Table 6.** Dimensions and dynamic load of an angular contact ball bearing.

| d, mm | D, mm | B, mm | Dyn. Load, kN |
|-------|-------|-------|---------------|
| 10    | 30    | 9     | 5.4           |

**Table 7.** UNSM treatment parameters.

| Amplitude (μm) | Load (N) | Rotational Speed (rpm) | Feed (mm/rev) |
|---|---|---|---|
| 30 | 30 | 30 | 0.07 |

**Table 8.** Bearing fatigue test conditions.

| Rotation Speed, rpm | Load, kN | Maximum Contact Pressure, MPa | Vibration Limit, G | Lubrication |
|---|---|---|---|---|
| 1000 | 3.5 | 2.764 | 22 | ISO VG 46 |

## 3.2. Bearing Fatigue Test Result

Figure 12 shows the SEM images of the untreated and UNSM-treated specimens before the fatigue test. Table 9 shows the change in surface roughness and hardness of the raceway before and after UNSM treatment. The surface roughness of the untreated and UNSM-treated specimens was found to be 0.052 μm and 0.033 μm, respectively. The hardness of the untreated and UNSM-treated specimens was found to be 61.5 HRC and 64.2 HRC, respectively. The surface morphology of the untreated specimen showed polishing traces (see Figure 12a), while UNSM treatment eliminated those polishing traces and instead formed a micro-dimpled surface as shown in Figure 12b. As a result, it was confirmed that the friction and lubrication effect in the bearing can be maximized through the dispersion of stress concentration and oil pocket effect when the micro-dimpled surface is formed with low surface roughness. Micro-dimples serve as hydrodynamic bearings to provide additional lift and they also provide easier escape of wear debris to minimize the third-body abrasion [8]. The fatigue test results of the angular contact ball bearings are listed in Table 10. The fatigue life of the untreated and UNSM-treated angular contact ball bearings was $11.176 \times 10^6$ and $14.527 \times 10^6$ cycles (a fatigue life improvement of approximately 29.9%).

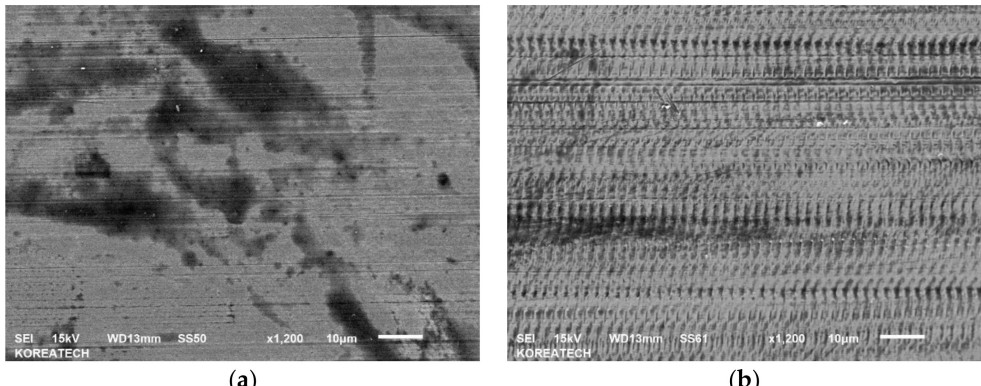

(**a**)            (**b**)

**Figure 12.** SEM images of the untreated (**a**) and UNSM-treated (**b**) specimens.

**Table 9.** Comparison of the surface roughness and hardness of the specimens.

| Specimens | Hardness, HRC | Roughness, μm, Ra |
|---|---|---|
| #1ACB untreated | 61.5 | 0.052 |
| UNSM-treated | 64.2 | 0.033 |

**Table 10.** Comparison of the fatigue cycles of the untreated and UNSM-treated specimens.

| Specimens | Cycles | Cause |
|---|---|---|
| #1ACB Untreated | 11,175,600 | inner ring raceway crack |
| #2ACB UNSM-treated | 14,527,800 | cage breakage |

The SEM analysis was performed to observe an insight into the failure mechanism. Figure 13 shows that cracks occurred in the inner race of the untreated specimen under repeated stress during the rotary contact fatigue test. On the other hand, the rotary contact fatigue test of the UNSM-treated specimen was stopped due to cage breakage. In Figure 13a, it can be seen that the crack width is 1.400 mm and the length is 1.203 mm. Figure 13b confirms that the crack has started between turning and polishing, while Figure 13c shows that the crack is distributed on the raceway.

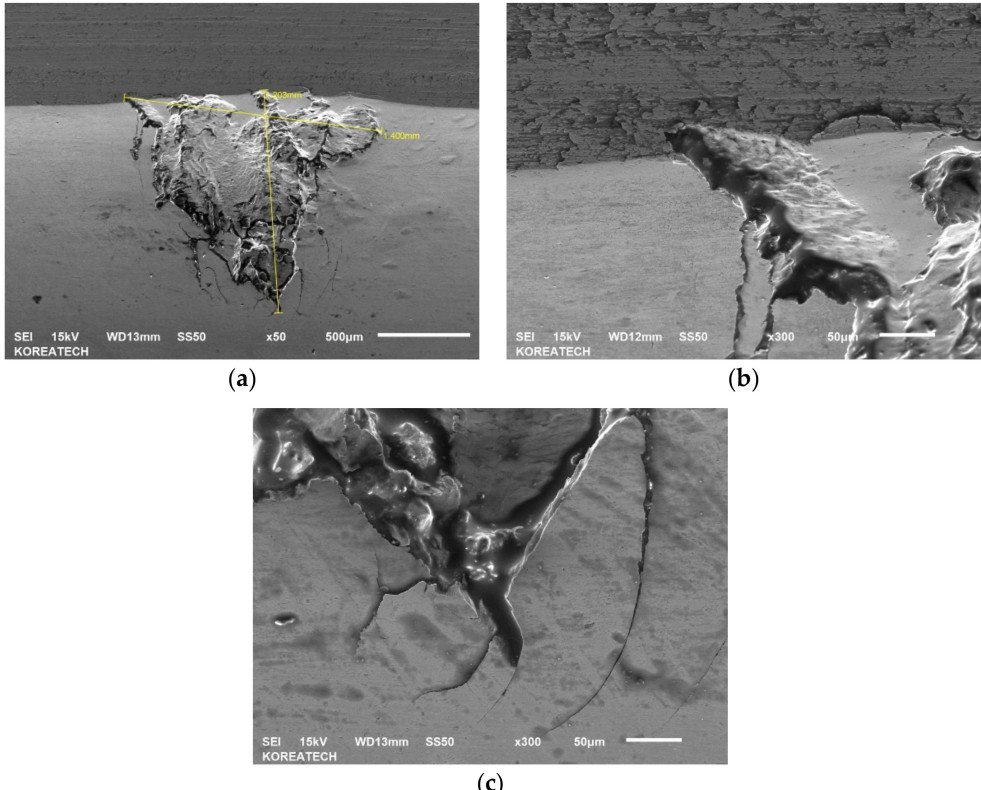

**Figure 13.** Comparison of low (**a**) and high (**b,c**) SEM images of the damaged untreated specimens (11.175 × 106 cycles). Testing conditions: Equivalent stress −1302 MPa, rotation speed −1000 rpm.

## 4. Conclusions

It this study, the effect UNSM treatment on the fatigue life of bearings was investigated and the following conclusions may be drawn:

1. The surface roughness of the UNSM-treated bearings was reduced from 0.550 μm to 0.149 μm, while the surface hardness of the UNSM-treated bearings was increased from 58 HRC to 62 HRC.
2. The improvements in the friction coefficient and wear resistance are considered to be the effect of surface roughness reduction, an increase in surface hardness, and also micro-dimples.
3. The longest fatigue life at 1484 MPa contact stress was found to be a 70.1% increase in the fatigue life of the UNSM-treated bearing compared to the untreated bearing.
4. The fatigue life of the untreated and UNSM-treated angular contact ball bearings was $11.176 \times 10^6$ and $14.527 \times 10^6$ cycles (a fatigue life improvement of approximately 29.9%).
5. Induced compressive residual stress by UNSM treatment played a dominant role in improving the fatigue life of bearings.
6. As a future work, the effect of UNSM treatment temperature on the fatigue life of bearings will be investigated. Also, a finite element analysis that can simulate the bearing working conditions needs to be performed.

**Author Contributions:** Conceptualization, Y.-S.P. and A.A.; methodology, S.D.; formal analysis, A.A. and S.D.; investigation, S.D.; data curation, project administration, J.-H.P.; A.A.; writing—original draft preparation, S.D. and Y.-S.P.; writing—review and editing, A.A.; supervision, J.-H.P., A.A. and Y.-S.P.; funding acquisition, A.A. and Y.-S.P.

**Funding:** This study was also partially supported by the Korea Technology and Information Promotion Agency (TIPA) for Small and Medium Enterprises. Project (No. S2544322)". "This study was also supported by the Industrial Technology Innovation Development Project of the Ministry of Commerce, Industry and Energy, Rep. Korea (No. 10067485).

**Conflicts of Interest:** The authors declare no conflict of interest.

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
