# Peer review of "A Study on the Improvement of the Fatigue Life of Bearings by Ultrasonic Nanocrystal Surface Modification Technology"

_metals, doi:10.3390/met9101114_

Round 1

Reviewer 1 Report

The problem statement isn't clear at all.

The language needs extensive review.

The introduction is very poor, more recent and related references should be added. 

Adding a flow chart to describe the research stages is required. 

The provided results in section 3 "Angular Contact Ball Bearing Fatigue Life Test" MUST be explained with more physical analysis.

The quality of almost all figures need to be improved.

The novelty of this work is still not clear and adding a new paragraph to highlight that is essential. 

The conclusions section needs to be re-written in an attractive way rather than the current one.

More details should be added about the future work. 

Author Response

We would like to thank the reviewers for the time and effort to review the manuscript and provide me with valuable comments and suggestions. The paper has been revised to the best of our ability and knowledge based on all the comments of the reviewers. The point-by-point responses to the reviewers` comments have been described in a separate document. In addition, all the revisions have been highlighted by a red font in the revised manuscript.

Reviewer 2 Report

This manuscript reports on the effect of UNSM technology that modifies the surface of metals (residual compressive stress, smaller grains hence larger yield strength) on fatigue life of rolling bearings. Overall, the results are clear and interesting.

The following should be clarified before publication

Page 5 line 113: 2.1 or 21 times? There is likely a typo in table 5.

Figures 8, 9, and 14: what are the testing conditions? A clearer explanation of the observed features, and their comparison, is required.

Figure 7: have repeats been done to get an idea of the error bars / spread in the fatigue life?

Last sentence of page 9 (“as a result, it was confirmed…”): how is that confirmed based on Figure 13? A better explanation is required.

There is a typo in table 10. 11,175,600 cycles.

Author Response

(The authors gave the same response as above.)

Round 2

Reviewer 1 Report

Accept in the current form.